# Probing a battery electrolyte drop with ambient pressure photoelectron spectroscopy

Julia Maibach [1,2], Ida Källquist [3], Margit Andersson[4], Samuli Urpelainen [4], Kristina Edström[1], Håkan Rensmo[3], Hans Siegbahn[3] & Maria Hahlin [3]

Operando ambient pressure photoelectron spectroscopy in realistic battery environments is a key development towards probing the functionality of the electrode/electrolyte interface in lithium-ion batteries that is not possible with conventional photoelectron spectroscopy. Here, we present the ambient pressure photoelectron spectroscopy characterization of a model electrolyte based on 1M bis(trifluoromethane)sulfonimide lithium salt in propylene carbonate. For the first time, we show ambient pressure photoelectron spectroscopy data of propylene carbonate in the liquid phase by using solvent vapor as the stabilizing environment. This enables us to separate effects from salt and solvent, and to characterize changes in electrolyte composition as a function of probing depth. While the bulk electrolyte meets the expected composition, clear accumulation of ionic species is found at the electrolyte surface. Our results show that it is possible to measure directly complex liquids such as battery electrolytes, which is an important accomplishment towards true operando studies.

[1] Department of Chemistry – Ångström Laboratory, Uppsala University, Box 538, 751 21 Uppsala, Sweden. [2] Institute for Applied Materials (IAM), Karlsruhe Institute of Technology (KIT), Hermann-von-Helmholtz-Platz 1, 76344 Eggenstein-Leopoldshafen, Germany. [3] Department of Physics and Astronomy, Uppsala University, Box 516, 751 20 Uppsala, Sweden. [4] MAX IV laboratory, Box 118, 221 00 Lund, Sweden. Correspondence and requests for materials should be addressed to J.M. (email: julia.maibach@kit.edu) or to M.H. (email: maria.hahlin@physics.uu.se)

Developing and implementing of renewable energy sources, and efficient energy storage solutions are prerequisites for creating a sustainable, fossil fuel-free society. Rechargeable batteries, such as lithium ion batteries (LIBs), currently dominate as portable energy sources in electric vehicles, computers, and mobile phones. Here, it is recognized that the solid-electrolyte interphase (SEI)[1,2] on the negative electrode plays a vital role for the long-term stability and safety of LIBs, and much research is focused on understanding its formation, composition, and functionality.

Investigations of the SEI commonly use scanning electron microscopy (SEM)[3–5], transmission electron microscopy (TEM)[6,7], photoelectron spectroscopy (PES)[4,5,8], and Fourier transform infrared spectroscopy (FTIR)[5,6], along with diffraction[4,7] and electrochemical methods[3,6,7]. Although these diverse methods yield complementary information about the structure, composition, functional groups, and resistivity of the SEI, the full characteristics of its formation remain unclear. This indicates the complexity of the system[9].

Since many crucial reactions in a battery occur at the electrode surface, PES with its surface sensitivity of a few nm and its element and compound sensitivity is one of the most suitable techniques to investigate battery interfaces[8]. Synchrotron radiation with both higher and lower photon energies than conventional lab sources has enabled such non-destructive depth characterization of the SEI[10,11]. The use of high-energy X-rays also allows one to probe the underlying electrode material and its lithiation mechanisms.

To date, however, PES has had one severe shortcoming with respect to battery characterization: the need for high vacuum conditions during measurements. Consequently, traditional experimental conditions cannot perfectly mimic the real battery, since they lack the liquid phase. Because important information may be unattainable when investigating dry samples post mortem, the ability to obtain in-situ measurements under conditions resembling the battery environment would represent a key development towards gaining a deeper understanding of this interphase and promoting further advances in LIBs.

Luckily, continuous method development has diminished the high vacuum constraint. Today's instruments can measure PES spectra far above the vapor pressure of water, and many other technologically relevant liquids[12,13] by using small apertures and efficient differential pumping systems[14]. This allows one to measure liquid electrolytes and their interactions with the electrode material under more realistic conditions. Ambient pressure photoelectron spectroscopy (APPES) equipment is available both at synchrotron facilities[15–17] and on-site at some institutes worldwide[18–20]. At synchrotron facilities, the high brilliance radiation gives additional advantages, such as, high resolution and a tunable photon energy ranging from soft to hard X-rays[11] that permits for depth profiling.

Although APPES has already been used to investigate catalysis[21,22], energy materials[23,24] and environmental chemistry[25,26]; most studies have targeted solid-gas interfaces[27,28] and reports on solid-liquid interfaces have been scarce[12,29,30]. The latter studies, as well as, recent work targeting the electrochemical double layer[31,32] are based on a dip-pull approach using aqueous electrolytes. The wide interest of understanding solid/liquid interface phenomena in diverse fields of science shows the need for method developments towards more realistic measurement conditions. A prerequisite to perform these interface studies is to first understand the liquid/gas interface for correct data interpretation, and this technologically important question is addressed in the current work.

In this study, we use alkyl carbonate-based electrolytes common in LIBs and a different approach to prepare the solid-liquid interface. In a previous study on solid-liquid interfaces of LIBs using APPES, we developed a new sample transfer protocol to transfer sensitive samples such as battery electrodes without air exposure directly from ambient pressures of a controlled environment to the measurement chamber without an intermediate high vacuum step[33]. This procedure does not require dedicated tubes/pipes for electrolyte inlets as for example liquid-jet experiments. This allows one to easily introduce a complex liquid droplet (e.g., a realistic battery electrolyte at 1M salt concentration) into the analysis chamber. If optimal conditions are achieved, one can probe the solid surface, the liquid, and the gas phase simultaneously in one spectrum. Maintaining a liquid electrolyte in the analysis chamber and using a set-up with dedicated electrical feedthroughs make operando APPES measurements now a realistic target.

Recognizing the prospect of future operando APPES measurements, we here identify and take a necessary step in the required methodology development. Traditional PES experiments of battery surfaces are complex due to highly reactive and radiation sensitive materials. On top of that, system-inherent peak shifts caused, for example, by the state of charge and surface layer formation are commonly observed[34,35]. Moving to APPES with a liquid electrolyte present will further complicate experimental considerations and interpretations, as the top electrolyte layer will dominate the signal.

Therefore, this article provides a study of a liquid carbonate-based battery electrolyte consisting of 1 M bis(trifluoromethane) sulfonimide lithium salt (LiTFSI) in propylene carbonate (PC) in comparison with the bare electrolyte solvent (PC) using ambient pressure conditions of PC gas. This choice of ambient conditions leads to more stable measurement conditions for battery electrolytes in APPES. We present results on the electrolyte composition and find concentration variations of its components with an accumulation of ionic species compared to solvent molecules in the surface region. We identify the importance of such surface concentration gradients in the liquid phase, the choice of ambient gas and further technical aspects such as radiation exposure as a prerequisite for reliable future operando APPES measurements of solid/liquid interfaces in general and battery experiments in particular.

## Results

**Separating salt and solvent signals in APPES spectra.** APPES measurements were performed on two samples: a solvent drop, consisting of pure PC on Li metal and a liquid electrolyte drop, consisting of 1M LiTFSI in PC on Li metal. These samples are hereafter referred to as Solv-Drop and Elect-Drop, respectively. The flexible Li substrate was bent up at the lower edge to support the position of the liquid droplets. This led to a liquid droplet thickness in the range of several mm (Fig. 1a). This thickness is several orders of magnitude larger than the PES probing depth. An ambient gas of PC at a pressure of 0.2 mbar was used in the high-pressure cell of the SPECIES end station at the MAX IV synchrotron. The ambient PC gas was found essential for maintaining a stoichiometric stable electrolyte drop (Supplementary Fig. 1 and Supplementary Note 1). It was also checked that the ambient PC gas and substrate made no contribution to the obtained spectra. Furthermore, measurements of a pristine substrate and the same substrate after PC vapor exposure showed no indication for a chemical reaction between the substrate and the PC vapor. From this, we conclude that the substrate surface is non-reactive (Methods section on sample preparation and Supplementary Fig. 2).

The APPES spectra of the Solv-Drop sample are shown in Fig. 1 (top, black spectra) and corresponding binding energy

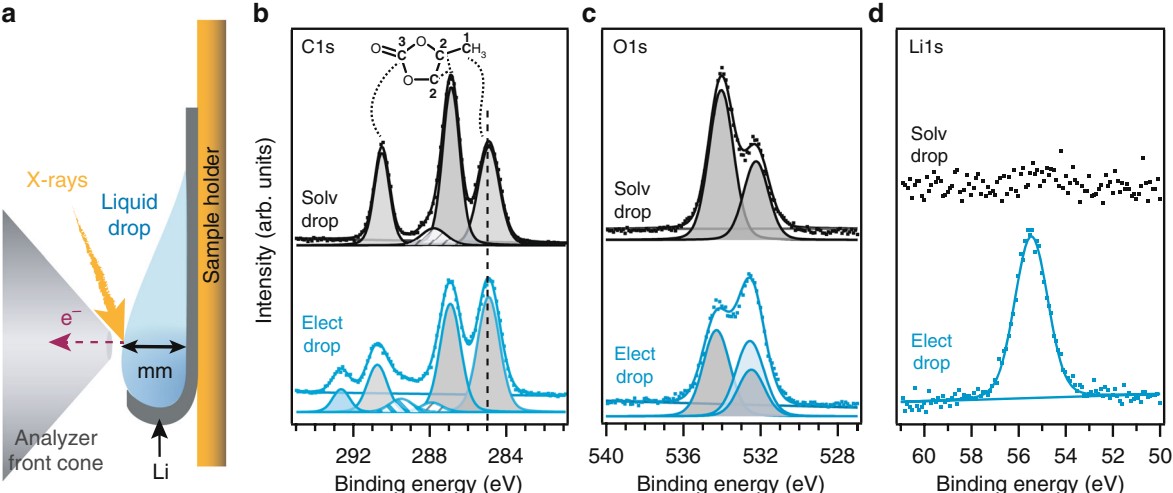

**Fig. 1** APPES measurements of PC solvent and 1 M LiTFSI in PC electrolyte. Schematic side view representation of the respective liquid droplets on the Li substrate (**a**). C 1s (**b**), O 1s (**c**), and Li 1s (**d**) PES spectra of Solvent drop (Solv-Drop, top), and Electrolyte drop (Elect-Drop, 1 M LiTFSI in PC, as prepared, bottom)

**Table 1 Overview of assigned peaks and their respective binding energy positions in the curve fits of C 1s, O 1s, and Li 1s spectra from Solvent drop (Solv-Drop), and Electrolyte drop (Elect-Drop)**

|  |  | Solv-Drop (eV) | Elect-Drop (eV) |
|---|---|---|---|
| C 1s | CH/PC (**C1**) | 285.0 | 285.0 |
|  | C-O/PC (**C2**) | 287.0 | 287.0 |
|  | C=O/O-C-O | 287.9 | 288.0 |
|  | C-F$_x$ | – | 289.4 |
|  | CO$_3$/PC (**C3**) | 290.6 | 290.5 |
|  | CF$_3$ | – | 292.6 |
| O 1s | PC C-O-C | 532.3 | 532.5 |
|  | PC C=O | 534.1 | 534.3 |
|  | LiTFSI | – | 532.6 |
| Li 1s | Li$^+$ | | 55.5 |

values obtained in the curve fit can be seen in Table 1. The binding energy scale is calibrated versus the hydrocarbon peak at 285 eV (further information in data treatment section). The absence of signal in the Li 1s region supports the visual assessment that the drop of solvent at the measurement spot is thick enough to prevent any signal from the underlying substrate. The absence of a Li peak also shows, that there is no dissolution of Li from the substrate into the solvent. Thus, it is concluded that only PC contributes to the signal and that the substrate does not contaminate the surface of the drop. The C 1s spectrum shows three large peaks at binding energies of 285.0 eV (C1), 287.0 eV (C2), and 290.6 eV (C3) in relative intensities (1.5:2:1.1) close to the expected stoichiometric ratio of 1:2:1 (with the exception of a slightly over-represented hydrocarbon component). In addition, a minor contribution at 287.9 eV is observed. This binding energy may be expected for a carbon in carbonyl group or a carbon-bonded to a single fluorine atom, however, the origin of this component is still unknown. The O 1s spectrum of the Solv-Drop sample shows two main features at binding energies 532.3 eV and 534.1 eV in a relative intensity ratio of ~1:2. Combined, the O 1s and C 1s spectroscopic signature fits very well with the one expected from the molecular structure of the PC molecule, three chemically different carbon positions and two chemically different oxygen positions in relative numbers matching the observed ratios in the APPES spectra. To the best of

our knowledge, these are the first spectra of PC in liquid phase. The stoichiometry of PC is well reproduced, and the data quality seems to surpass that of previous reports on solid/frozen PC[36].

The APPES spectra of the Elect-Drop sample are shown at the bottom of Fig. 1, with binding energy values obtained from the curve fit summarized in Table 1. The additional peaks in the C 1s, O 1s, and Li 1s spectra for the Elect-Drop sample (compared to the Solv-Drop sample) originate from the LiTFSI salt. The salt has two carbon atoms in identical S-**C**-F$_3$ chemical environments. In the C 1s spectrum, this peak is clearly observed at 292.6 eV. In the O 1s spectrum, the increased intensity at 532.4 eV is in line with the oxygen in the SO$_2^-$ units in the TFSI-anion. Moreover, the Li 1s spectrum shows an intense lithium peak at a position similar to that of ionized lithium.

In the C 1s spectrum of the Elect-Drop sample, the PC attributed peaks can be observed, although the carbonate component appears at a slightly lower-binding energy (0.1 eV lower). In addition, the intensity ratio between C3 and C2 in PC (see Fig. 1) changes from 1.1:2 for the pure PC solvent to 0.83:2 for PC in the electrolyte. These changes could indicate a surface organization of the PC molecules, where the C1 and C2 carbon species of PC are on average closer to the surface of the Elect-Drop than C3. This seems in line with simulation results of Skarmoutsos et al. showing that solvated Li ions will be coordinated preferentially by the C=O moieties (C3) of the PC molecules in Li(PC)$_3^+$ or Li(PC)$_4^+$ clusters[37]. In such configurations, C2 and C1 would point towards the electrolyte surface leading to stronger signals in PES compared to C3. In addition, a more intense hydrocarbon peak at 285.0 eV is observed in the spectra for the Elect-Drop sample. This increased intensity is not expected from carbon sites in the salt anion. However, in view of a generally expected increase of the surface tension due to the addition of salt, an increase in adsorbed ambient hydrocarbons would be expected. Previous work using high humidity during APPES has shown an accumulation of hydrocarbons during the analysis[38], and our results indicate similar behavior for the PC. As for the Solv-Drop sample, a peak at approximately the carbonyl/C-F binding energy (288 eV) is observed also for the Elect-Drop sample. In addition, a peak at 289.4 eV is seen that could be explained by a decomposition of the CF$_3$ group from the TFSI anion to different C-F$_x$ species (Fig. 1).

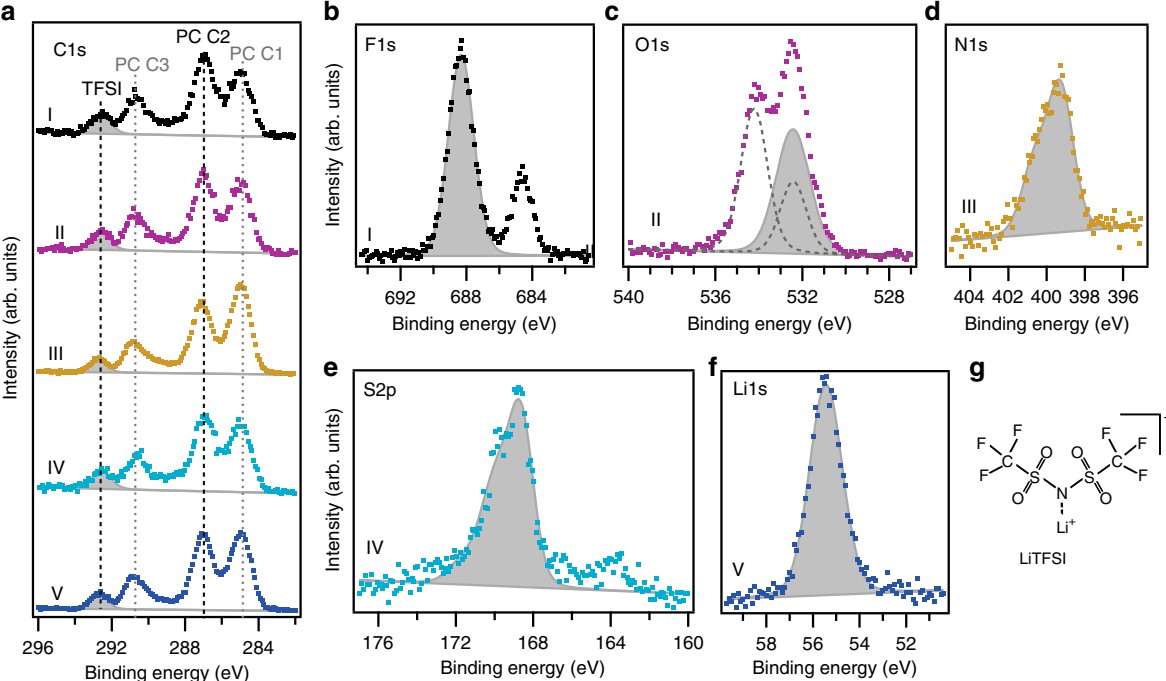

**Fig. 2** APPES data for composition analysis of 1 M LiTFSI in PC electrolyte. C 1s (**a**), F 1s (**b**), O 1s (**c**), N 1s (**d**), S 2p (**e**), and Li 1s (**f**) spectra of the Elect-Drop sample (1 M LiTFSI electrolyte on Li metal). The shaded areas represent the respective TFSI contributions in case of overlapping signals. The roman numerals indicate different measurement spots. The molecular formula of LiTFSI is shown in (**g**)

**Electrolyte composition—stoichiometric calculations**. To confirm the electrolyte composition in the droplet, relevant peak areas were evaluated. In a first step, the solvent-to-salt ratio within the probed volume is determined. Therefore, all C 1s spectra taken for the Elect-Drop sample (Fig. 2a) were curve-fitted to separate the different contributions of solvent (PC C1, C2, and C3 according to inset in Fig. 1) and salt. Subsequently, the areas of the fitted peaks corresponding to PC (C2, ether C–O at 287 eV as indicated by the dashed line) and TFSI-anion (CF$_3$ at 292.6 eV) were calculated. The ratio between the respective peak areas then gives an estimate of the number of PC molecules per TFSI-anion. This evaluation gives a ratio of approximately 5–6 PC molecules per TFSI-anion. The detailed results are summarized in Supplementary Note 2 and Supplementary Table 1. Based on the molecular weight of PC (102.09 g mol$^{-1}$, CAS number 108-32-7) and its density (1.204 g cm$^{-3}$), the molar ratio can be calculated. Thus, in a 1 M LiTFSI in PC solution, the molecular ratio between LiTFSI and PC is roughly 1:12. Therefore, our APPES results indicate a salt enrichment at the surface of the Elect-Drop sample equivalent to a surface layer concentration of around 2 M LiTFSI.

To further verify salt enrichments at the surface of the drop, a similar Elect-Drop sample was prepared and measured with an in-house APPES instrument using Al Kα radiation as excitation energy[33]. The corresponding C 1s spectrum is shown in Fig. 3 (black solid line). For comparison, also the spectrum measured at 835 eV excitation energy is added (blue dotted line) and both spectra are normalized to the C-O intensity at a binding energy of 287 eV. For the more bulk sensitive measurement (Al Kα) it is clearly seen that the relative ratio between TFSI (CF$_3$) to solvent C-O is lower, and when fitted showed a ratio of 1:11, i.e. close to the expected value of 1:12. These results confirm salt enrichment at the surface of the Elect-Drop sample. Note that determining salt enrichment is dependent on understanding and controlling the measurement conditions, since the drop stability, as well as

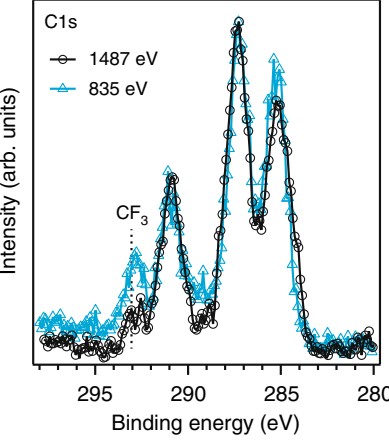

**Fig. 3** Varying APPES probing depth on 1 M LiTFSI in PC electrolyte drop. C 1s spectra of Elect-Drop samples using two excitation energies showing the increased CF$_3$ intensity relative to the PC related intensity for higher surface sensitivity (i.e. 835 eV excitation energy). The intensities are normalized with respect to the C-O peak at 287 eV

radiation sensitivity, may otherwise influence the results (Supplementary Fig. 3 and discussion in Supplementary Note 3).

To investigate the stoichiometry of the dissolved salt, the measurement spot was moved to a fresh location for each element in the salt to limit radiation damage effects. First, a carbon spectrum was recorded for energy calibration and intensity evaluation of the CF$_3$ peak, followed by the targeted elemental line (Fig. 2). Thus, the salt composition is evaluated with respect to the CF$_3$ peak in the respective C 1s spectra. This procedure requires a rather homogeneous electrolyte surface. In our experiments, we confirmed the homogeneity by comparing the

**Table 2 Evaluation of the relative amounts of the different LiTFSI components in the Elect-Drop sample**

| Element | Calculated ratio | Stoichiometric ratio from salt structure |
|---|---|---|
| C ($CF_3$) to F ($TFSI^-$) | 2: 8.4 | 2: 6 |
| C ($CF_3$) to O ($TFSI^-$) | 2: 3.8 | 2: 4 |
| C ($CF_3$) to S ($TFSI^-$) | 2: 1.5 | 2: 2 |
| C ($CF_3$) to N ($TFSI^-$) | 2: 1.6 | 2: 1 |
| C ($CF_3$) to Li ($Li^+$) | 2: 12.3 | 2: 1 |

C 1s spectra from the five different measurement spots. At two spots, larger amounts of hydrocarbons were observed, otherwise the relative ratio between salt and solvent is constant and can be used as reference for the salt stoichiometry analysis. Finding such a homogeneous electrolyte surface shows that decomposition products, observed during exposure to radiation (Supplementary Fig. 3), do not spread to nearby measurement spots within the time frame of the measurements. The variations in signal-to-noise-ratio are due to the instrument constraints, where not all sample positions can be fully optimized with respect to the maximum count rate and the available measurement time.

Besides carbon and oxygen (the spectra of which have already been discussed in the previous section), the salt contains fluorine, sulfur, and nitrogen. These spectra are shown in Fig. 2. In the F 1s spectrum, at least two different chemical environments can be found. Since the only source of fluorine is the $CF_3$-moiety of LiTFSI-salt, this implies some extent of salt decomposition. The intensity ratio between the two F 1s components is roughly 3:1, indicating breakdown of the $CF_3$-moiety in the salt. The $CF_3$ chemical environment is expected at the higher binding energy position (688.4 eV). The second component has a binding energy of 684.7 eV and most likely originates from solvated fluoride ions (based on the general assignment of this binding energy to LiF for solid samples). As a clear $CF_3$ peak can be detected in the C 1s spectrum, a large amount of the salt is still intact.

The main peaks in the N 1s region at 399.3 eV and in the S 2p region at 168.7 eV coincide reasonably well with the binding energies expected for the nitrogen and sulfur chemical environments in the TFSI anion[39]. In the S 2p spectrum, a second peak at around 163.8 eV also indicates reductive salt decomposition, as this binding energy is characteristic for sulfur in the oxidation state 0. In addition, the asymmetric shape of the N 1s peak further supports that some of the TFSI-ions have degraded. The determination of the exact decomposition mechanism of the salt is beyond the scope of this article but agrees with the previously postulated LiTFSI decomposition routes[40,41] and reported (electro)chemical instability of $MgTFSI_2$[30].

The salt stoichiometry evaluation is summarized in Table 2. It is based on the intensities that can be attributed to the intact LiTFSI salt (i.e., not including intensity that can clearly be attributed to decomposition products). Therefore, the F 1s, O 1s, S 2p, N 1s and Li 1s spectra were curve fitted and the area of the relevant peaks are used for calculating the ratio between different elements in the salt. The peak areas included for the stoichiometry calculation are shaded gray in Fig. 2. The detailed description of the calculation together with the input parameters such as integrated areas, cross sections from ref. [42], calculated inelastic mean free paths are presented in Supplementary Note 4. To compare the results of this intensity evaluation directly to the stoichiometric formula of LiTFSI (Fig. 2g), $n_{CF3}$ is set to two.

There is a clear discrepancy between the calculated and expected ratios from the stoichiometric formula for all elements.

The deviation for most elements is in the order of up to 50%. However, we found 10 times more lithium in the probed volume than expected. This exceeds the other deviations from the expected salt composition by far and we, therefore, consider this relevant. It shows qualitatively that we have a significant surface enrichment of the Li ions in the Elect-Drop sample.

Since we could not observe dissolved Li in the Solv-Drop sample, we believe that this surface enrichment is based on a concentration gradient in the electrolyte surface. Combining the increased Li intensity with the observed fluoride ion peak at a binding energy of 684.7 eV, we evaluated to what extent $F^-$ ions could compensate for the excess $Li^+$ intensity. Based on normalized intensities and known atomic ratios of F and Li in both LiTFSI and LiF, we estimate that only about 10% of the Li 1s intensity should stem from Li ions that have $TFSI^-$ as counter ion and 70 % could be explained by $Li^+$ with fluoride ($F^-$) as counter ion (see Supplementary Note 5 for detailed description of this evaluation). The remaining intensity could stem from lithium carbonate type compounds and related solvated Li ions.

## Discussion

In this article, a drop of solvent (propylene carbonate) and a drop of electrolyte (1 M LiTFSI in propylene carbonate (PC)) were investigated with ambient pressure photoelectron spectroscopy (APPES) to provide relevant insights for future operando experiments on solid/liquid battery interfaces.

For the first time, we recorded high quality photoemission spectra of liquid propylene carbonate. It was possible to stabilize the drop of PC solvent and the drop of electrolyte using PC gas at its vapor pressure of 0.2 mbar as the background gas during the measurement. This is in contrast to measurements obtained in 2 mbar ambient pressure of Ar gas, which resulted in the instantaneous evaporation of the PC drop and a continuous evaporation of PC from the electrolyte drop.

Our results show that the presented approach enables direct APPES studies of electrolyte solutions with battery-relevant concentrations. This is an important accomplishment toward operando studies involving complex liquids. Under the much more stable measurement conditions achieved in this study, we were able to further evaluate the stoichiometry and surface composition of the electrolyte. The electrolyte composition obtained in the more depth sensitive measurement (using 1487 eV excitation energy) was closer to the expected stoichiometry, while the more surface sensitive measurement showed higher salt concentration in the probed volume. This indicated a salt enrichment at the surface of the Elect-Drop sample, since the molar ratio gave a relative concentration of PC:LiTFSI of approximately twice that found in the bulk.

Based on our findings, we believe that strategies for future in-situ and truly operando battery experiments will have to include the systematic characterization of the electrode substrate, the liquid phase (both pure solvent and mixed electrolyte), and the gas phase used to stabilize the liquid. Also, optimal conditions for probing a buried interface require understanding the trade-off between interface sensitivity and probing depth. Considering the high surface sensitivity of photoelectron spectroscopy, surface concentration gradients need to be accounted for.

From our static droplet experiments, we conclude that, even without applying any potential to the electrode, adding the liquid phase to the electrode surface increases the complexity of PES studies drastically. A correct interpretation of in-situ and operando APPES battery experiments can only be achieved when we fully understand the behavior of the liquid electrolyte under the given conditions.

## Methods

**Materials.** Unless otherwise specified, all chemicals were handled under inert argon atmosphere inside a glove box ($H_2O$ ~1 ppm, $O_2$ ~1 ppm) or a glove bag. To introduce the samples into the APPES instrument we used a glove bag, which was Ar-flushed for a minimum of four times directly before any new sample preparation. For chemical and sample transport, the materials, as well as vials of liquid solvent or electrolyte, were either sealed in individual vacuum pouch cells or stored in argon-filled KF-tubes to avoid all contact to atmospheric conditions.

The electrolyte was prepared by dissolving 1M bis(trifluoromethane) sulfonimide lithium salt (LiTFSI, BASF, purity 99.9 wt%) in propylene carbonate (PC, BASF, purity 99.9 wt%). The salt was dried overnight at 120 °C under vacuum. The solvent PC was used as received.

In addition, 10 ml PC was degassed by placing PC in a vessel with a small opening in a vacuum chamber (pressure ~$2 \times 10^{-3}$ mbar, using a backing pump) and a Teflon piece, as gas nucleation help. The low pressure was applied until no further gas evolution could be observed (~60 min). At MAX IV Laboratory, the degassed PC was transferred into a quartz test-tube connected to a CF16 flange with a Swagelok valve, which was subsequently connected to the APPES cell at the SPECIES end station. All gas lines were pumped to low vacuum before the valve to the PC test-tube was opened. Prior to use, the PC test-tube was pre-pumped to low vacuum conditions to remove the remaining Ar gas.

**Instrumentation.** Photoelectron spectroscopy characterizations were performed at the SPECIES beamline (MAX IV Laboratory, MAX II ring). The beamline is equipped with an APPES end station, which allows surface sensitive characterization of solid-liquid interfaces at pressures up to 25 mbar in the energy range of 27–1500 eV[43].

The end station is based on a cell concept, which allows both clean UHV conditions, as well as APPES measurements in a retractable high-pressure cell. The data presented here were recorded entirely using the high-pressure cell. Fixed excitation energy of 835 eV, and pass energies of 50 eV (Solv-Drop) and 100 eV (Elect-Drop) were used. In both cases, the spectral resolution was limited by the sample. The incident photon angle was 54 degrees, and the photoelectron emission angle was 90 degrees, both relative to the plane of the sample substrate. The beamline and end station are described in detail in refs. [16,43,44].

The spectra were measured in a PC atmosphere with a base pressure of ~0.2 mbar (vapor pressure of PC at 20 °C is 0.173 mbar). In all cases, a series of spectra were recorded at different time intervals and different exposure times to synchrotron radiation. For clarity of presentation, only one representative data set per sample environment is shown.

APPES measurements using higher excitation energies were performed at Scienta Omicron in Uppsala using an APPES set-up described in ref. [19]. The system is equipped with a monochromatic Al Kα source and a HiPP-3 analyzer using swift acceleration mode[45].

**Sample preparation.** For the synchrotron experiments, lithium metal (Cyprus Foote Mineral, 125 μm thick) was mounted on the sample holder using conductive Cu tape inside an argon filled glovebox. The lower edge of the soft Li substrate was bend upward to provide an edge to support the liquid drops.

The preparation of a drop sample was done by placing a solvent (PC) or electrolyte (1 M LiTFSI in PC) drop on the lithium foil, following the sample preparation according to ref. [33]. Once the samples with the liquid drop were placed inside the APPES setup, the valve to the PC-filled test-tube was opened and the cell was evacuated to 0.2 mbar equal to the PC vapor pressure. During the measurements, the samples were held in an atmosphere of PC gas at its vapor pressure. The measurement positions were chosen deliberately on the thick parts of the adsorbed drops to avoid contributions for the substrate and study purely the liquid phase.

For reference measurements using an in-house APPES set-up, lithium metal was mounted on a sample holder inside an Ar-flushed glove bag and a drop of electrolyte was added when the sample holder was in place on the manipulator following ref. [33]. The pressure in the analysis chamber was held at 2 mbar using Ar gas.

**Data treatment.** All data evaluation including curve fitting was performed using the Igor Pro 6.34 software package. The spectra were calibrated in binding energy with respect to the hydrocarbon peak at 285.0 eV, which is a procedure that has often been used in the battery context. The spectra of the pure solvent (Solv-Drop) were fitted with the minimum number of required peaks. The obtained binding energy differences were then used as input parameters to fit the electrolyte drop spectra together with new peaks related to the TFSI anion. Only in case of the Elect-Drop O 1s spectrum, a more complex approach was needed due to the strong overlap of the different oxygen species. Therefore, the TFSI O 1s component was firstly determined from a difference spectrum and then kept fixed during the subsequent curve fitting procedure.

## Data availability

All relevant data are available from the authors upon request.

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

## Acknowledgements

The authors thank for generous sponsoring of this work from the Swedish Research Council (2016-03545, 2012-4681, 2014-6019, 2018-06465), ST and UP for Energy, the Swedish Energy Agency (40495-1), and the Carl Trygger Foundation. In addition, J.M. gratefully acknowledges funding from the German Federal Ministry of Education and Research (FKZ 03XP0131). The authors would also like to thank the staff at MAX IV laboratory for friendly support during measurements, Henrik Eriksson (Uppsala University) for experimental support in Uppsala and Andrew J. Naylor for valuable input. Finally, we would like to thank Scienta Omicron for the possibilities to perform APPES measurements at their facilities and in particular acknowledge John Åhlund for experimental support and valuable discussions.

## Author contributions

J.M. and M.H. designed the experiments and analyzed the data. J.M., I.K., and M.H. prepared the samples for measurements and performed the photoelectron spectroscopy measurements together with M.A. and S.U. J.M., I.K., K.E., H.R., H.S., and M.H. interpreted and discussed the data. J.M., I.K., and M.H. wrote the article with valuable input from K.E., H.R., and H.S.

## Additional information

**Competing interests:** The authors declare no competing interests.

