## [Peer Review File · Nature Communications]

Reviewers' comments:

Reviewer #1 (Remarks to the Author):

The authors presented the operando ambient pressure x-ray photoelectron spectroscopy (AP-XPS) characterization of a model electrolyte based on 1 M bis(trifluoromethane)sulfonimide lithium salt (LiTFSI) in propylene carbonate (PC), using PC vapor as the stabilizing environment. The approach and the results are essential and interesting. However, some concerns need to be clarified before it is acceptable for publication.

(1) When the PC solvents contact with Li, PC is suggested to be decomposed and a solid electrolyte interphase (SEI) is formed on the Li surface. The result-“ there is no dissolution of Li from the substrate into the solvent”, is unclear to the reviewer. Did the authors check the surface of Li in the Solv-Drop sample? Is any hint from the carbonyl peak at 287.9 eV?

(2) The authors claims that a salt enrichment at the surface of the Elect-Drop sample is equivalent to a surface layer concentration of around 2 M LiTFSI. However, how to differentiate a SEI environment and near-surface solution environment?

Reviewer #2 (Remarks to the Author):

This is an interesting battery related surface science work with unique procedure of using PC solvent vapor pressure instead of traditional Ar partial pressure for AP-XPS measurements. Although the method is unique, the results and discussion are not strong and authors acknowledge that fact as such. Nevertheless such a technique focused manuscript may be more suitable for field specific journal rather than Nature Commun. In my view the following queries are critical and needs to be properly explained.,

1. What is the size of the drop let., considering surface sensitive nature of XPS., did authors measure the top layer of the droplet or Li-metal and droplet interface?. Is the wettability of Li-metal was an issue? Did authors try to measure at the edge of droplet (such that X-ray is hitting 50% of the droplet and another 50% on Li-metal).

2. What about the surface contaminants of pristine Li-metal surface. Often they have organic contaminants which can mimic the reacted products? Did authors choose a bare Li-metal spot (assumably from one corner where there is no droplet?), this should be there background for data analyzing.

3. Authors detected LiF and CF₃ peak in F1s, but surprisingly no CF₂, or CF peaks (which are intermediate to form LiF).

4. LiF is insoluble in PC solvent, but the presence of LiF peak in F1s spectra indicates, authors are observing the metal-droplet interface?

5. small amount of elemental sulfur (~163eV) in S2p spectrum is interesting, but again, looks more like contaminant issue., although author claims this is not focus of the manuscript, these critical observations need to be analyzed (atleast provided hypothetical explanations)

6. Authors observe decomposition of CF₃ (as LiF) and SO₂ (as elemental sulfur) but no extra peaks observed in Nitrogen spectra?., If the TfSI molecule is decomposing, isn't Nitrogen from TfSI will also

form byproducts?

7. Authors need to identify and discuss more about C1s peak around 291eV (is this CF₂/CF)?

8. Another concern is the results vary with different measurement spots, there is no convincing results for such an inhomogeneous sample/measurement conditions.

Reviewer #3 (Remarks to the Author):

Reviewer report on:

Manuscript Number: Nature Comm.

NCOMMS-18-33746

Authors: Maibach et al.

Title: The surface of a battery electrolyte drop seen through ambient pressure x-ray photoelectron spectroscopy

General:

At first a personal comment: I am not a regular reader of "Nature Comm.". After inspecting some of the papers in the current field, I feel that most of the published papers have a more general message than the present one. This is my main doubt and thus you should include your editorial position and the opinion of the other reviewers into the final decision.

Nevertheless, the paper is dealing with a very actual challenging field: the improvement of the possibilities for storage of electrical energy efficiently in batteries for future use of renewable energy. The characterization of the complex chemical processes in the batteries is for that aim of existential importance. For more realistic analysis the closure of the "pressure gap" is of main interest. The experiments of the authors here are really at the forefront in the community.

The experiments, results and conclusions are convincing (with some exceptions - see remarks). To learn about the electrolyte/salt behavior is of high importance and the concept of the authors is valid. Main result is that XP measurements at liquid PC-based electrolyte is reliable possible and that demixing of electrolyte components can be proved. The study is a prerequisite for real electrochemical in-operando experiments, but it is still some steps away from it. And, I feel, in the "background" section the reader is given a little bit too much hope for such studies.

Detailed remarks:

If you decide for publication I have to add some specific remarks:

I am also not a native English speaker, but - in some parts the English has to be improved! One example - last sentence of conclusions: "Only if one understands how the liquid electrolyte behaves under AP-XPS condition can a correct interpretation of in-situ and operando battery experiments be achieved."

All abbreviations have to be defined? In the abstract already or in the main text? E.g. AP-XPS is explained in line 2, not in line 47, once more explained in line 57

The paper should be clearly readable also for non-expert readers, the mixture of PES and XPS here is not really necessary.

line 19: fossil-free "society" - what is this...?

l 63: what means "tubing" here

+ [33] if I understood the paper right there Maibach et al. already measured XP spectra of a drop of PC - thus it is not really the "first time" here...

l 65: 1 M concentration - of what

l 67-70: "using a set-up with dedicated electrical feedthroughs make operando AP-XPS measurements now possible" - this makes the reader wrong hope, not shown in the current paper!

I 80-81: "and the depth distribution of its components" - not really "depth distribution" - hints for demixing!

I 92 - comment that the "pristine" surface is Li-carbonate not Li metal !

from here energy values are discussed... a hint that the BE scale is calibrated (C1s 285 eV - see "data treatment") would here be helpful

I 108: "these are the first spectra of PC in liquid phase." ??? [33] see comment above

I 133, 140: 284.9 eV ??? BEs are calibrated to 285.0 eV!

concentration calculation - (S5) here a comment on the used procedure would be helpful..., error discussion?

I 176 177: "fresh location" - what this really means for a liquid droplet? I would assume always convection of material, at least at the surface...

I 195 very noisy peak !

I 208, I 217: table 2, discussion for C: Li = 2 : 12, mandatory! I cannot follow the discussion with mainly formation of LiF !! There is not enough excess of fluorine!

I 222-223: "The remaining intensity could stem from lithium carbonate type compounds and related solvated Li ions" - please, clearer statement on this

Proposal: what about making the same experiment with a substrate other than a Li foil? So one can rule out dissolution of Li...

I 240: "salt enrichment" OK, but not "depth distribution" as mentioned earlier in the paper - please, discuss HAXPES advantages here, too

I 264-266: what species are outgassing, water?,

how one can reach 2×10^{-3} mbar, if the vapor pressure of PC is 0.2 mbar; is there no evaporation of PC ???

I 290... please, explain the geometry (size) of the droplet

General reply to all reviewers:

From all reviewer comments, we identified that there is a strong need for us to further describe the sample geometry and give information about the thickness of the liquid droplets. We therefore included a schematic representation of the liquid droplet on the Li substrate in the manuscript in Figure 1.

We used the lithium substrate due to its flexible nature. This allowed us to bend the lower edge of the substrate upwards to give some support to the liquid droplet. This resulted in a liquid droplet thickness in the range of a few millimeters. Below we show two photographs of such a liquid droplet in the APPES measurement set-up but not in the measurement position. In the shown photographs, the liquid droplet is positioned below the analyzer front cone. The front cone tip diameter is just under 1 mm; thus, it can be estimated that the droplet has a thickness of 2-3 mm. Considering the short inelastic mean free path of electrons in matter, this is roughly one million times thicker than the PES probing depth. Thus, we are sure that we could not see the interface between PC and Li substrate.

These photographs were taken at the APPES set-up at Scienta Omicron, under similar measurement conditions as the measurements performed at the SPECIES beamline, but the droplet is not placed in the measurement position. Not to confuse the reader with this, a drawn figure was inserted in the manuscript instead showing a schematic side view of the set-up with the sample in measurement position.

Changes to the manuscript:

Updated Figure 1

p. 5 line 92: “These samples are hereafter referred to as Solv-Drop and Elect-Drop, respectively. The flexible Li substrate was bent up at the lower edge to support the liquid droplets. This led to an approximate liquid droplet thickness in the range of mm (see Figure 1 a). This thickness is several orders of magnitude larger than the PES probing depth.”

Figure 1 caption: Schematic side view representation of the respective liquid droplets on the Li substrate.

Reviewer #1 (Remarks to the Author):

The authors presented the operando ambient pressure x-ray photoelectron spectroscopy (AP-XPS) characterization of a model electrolyte based on 1 M bis(trifluoromethane)sulfonimide lithium salt (LiTFSI) in propylene carbonate (PC), using PC vapor as the stabilizing environment. The approach and the results are essential and interesting. However, some concerns need to be clarified before it is acceptable for publication.

(1) When the PC solvents contact with Li, PC is suggested to be decomposed and a solid electrolyte interphase (SEI) is formed on the Li surface. The result-“ there is no dissolution of Li from the substrate into the solvent”, is unclear to the reviewer. Did the authors check the surface of Li in the Solv-Drop sample? Is any hint from the carbonyl peak at 287.9 eV?

(2) The authors claims that a salt enrichment at the surface of the Elect-Drop sample is equivalent to a surface layer concentration of around 2 M LiTFSI. However, how to differentiate a SEI environment and near-surface solution environment?

Author reply:

In our view, the two remarks by reviewer 1 are linked and below follows our answer and our improvements to the article.

Prior to depositing the electrolyte drop, the lithium metal substrate was thoroughly investigated using PES. It was found that the pristine Li sample surface consisted of Li_2CO_3 and other compounds, and only a minor metallic Li 1s peak from the bulk lithium metal could be observed. The inertness of this lithium substrate was checked by exposing the lithium sample to PC gas, and no changes of the surface could be detected after exposure. The results of this investigation are included in Supporting Note 2. From these, we concluded that the Li used in this study was already passivated and did not react further with PC. This is already stated in the manuscript in line 100ff: “From this, we conclude that the substrate surface is non-reactive (see sample preparation and supporting information S2).”

During measurements the solvent drop had a thickness in the range of mm (see above general reply). Considering the short inelastic mean free path of electrons in matter, this is roughly one million times thicker than the PES probing depth. Thus, we are sure that we could not see the interface between PC and Li substrate. To further clarify this point in the manuscript we added a figure on the sample and measurement set up (see general reply above).

We also addressed the issue of possible contaminations from the substrate in the original manuscript now on page 5, line 106ff: “The absence of signal in the Li 1s region supports the visual assessment that the drop of solvent at the measurement spot is thick enough to prevent any signal from the underlying substrate. The absence of a Li peak also shows, that there is no dissolution of Li from the substrate into the solvent.” To clarify this point we modified the text.

Since no trace of lithium (or other dissolved compounds) could be detected on the surface of the pure PC drop, we interpret this as the compounds present on the lithium surface cannot be dissolved in the solvent/electrolyte. Taken both points, droplet thickness and lack of signal from lithium or dissolved compounds, into account, we concluded that we could fully omit the influence of the substrate in the following analysis.

Changes to the manuscript

p. 5 line 109: “Thus, it is concluded that only PC contributes to the signal and that the substrate does not contaminate the surface of the drop.”

Reviewer #2 (Remarks to the Author):

This is an interesting battery related surface science work with unique procedure of using PC solvent vapor pressure instead of traditional Ar partial pressure for AP-XPS measurements. Although the method is unique, the results and discussion are not strong and authors acknowledge that fact as such. Nevertheless such a technique focused manuscript may be more suitable for field specific journal rather than Nature Commun. In my view the following queries are critical and needs to be properly explained.

Author reply:

Apart from the large general interest in battery and energy storage, solid/liquid (e.g. corrosion, catalysis, biological cell membranes) and liquid/gas (e.g. aerosols in atmosphere, gas sensors) interface phenomena are of general importance to a broad research community in physics, chemistry, and biology. In light of this we believe that our results on real electrolytes are of common interest. To further highlight the importance of prerequisite studies such as ours to understand the liquid component, which is necessary to achieve correct data interpretation for these interface studies, the following changes were made in the introduction and we included liquid/gas interfaces as a key word.

Changes to the manuscript:

Added p. 4 line 56: “The wide interest of understanding solid/liquid interface phenomena in diverse fields of science shows the need for method developments towards more realistic measurement conditions. A prerequisite to perform these interface studies is to first understand the liquid/gas interface for correct data interpretation, and this technologically important question is addressed in the current work.”

Removed p. 4 line 61: ~~The operating pressure of these AP XPS instruments is higher than the vapor pressure of the most common electrolyte solvents used in LIBs allowing for a new type of measurements in this technologically important area.~~

1. What is the size of the drop let., considering surface sensitive nature of XPS, did authors measure the top layer of the droplet or Li-metal and droplet interface? Is the wettability of Li-metal was an issue? Did authors try to measure at the edge of droplet (such that X-ray is hitting 50% of the droplet and another 50% on Li-metal).

Author reply:

We have added a figure to illustrate the droplet size and geometry (see general reply above), and since the droplet is much larger and thicker than the beam spot and the probing depth, respectively, only the top layer of the droplet is measured. This is further confirmed by all points measured on the solvent drop looking very similar showing only PC features, with no signals from substrate species. We attempted to measure the interface between the liquid and the electrolyte at the top of the droplet where we knew from previous experiments that the liquid layer should be thin enough to allow access to the substrate/liquid interface. However, it was not possible to move the sample to such a position in the SPECIES set-up due to the physical constraints in sample positioning given by the experimental set-up.

The wettability of lithium was not an issue, but to make sure that the drop would not fall off the bottom edge of the lithium foil was curved upwards to support the droplet position, see new figure 1a.

Changes to the manuscript:

Schematic included in Figure 1.

p. 5 line 92: “These samples are hereafter referred to as Solv-Drop and Elect-Drop, respectively. The flexible Li substrate was bent up at the lower edge to support the position of the liquid droplets. This led to a liquid droplet thickness in the range of several mm (see Figure 1 a).”

2. *What about the surface contaminants of pristine Li-metal surface. Often they have organic contaminants which can mimic the reacted products? Did authors choose a bare Li-metal spot (assumably from one corner where there is no droplet?), this should be there background for data analyzing.*

Author reply:

A similar point was raised by reviewer 1. We therefore would like to refer to the same answer given to reviewer 1:

Prior to depositing the electrolyte drop, the lithium metal substrate was thoroughly investigated using PES. It was found that the pristine Li sample surface consisted of Li_2CO_3 and other compounds, and only a minor metallic Li 1s peak from the bulk lithium metal could be observed. The inertness of this lithium substrate was checked by exposing the lithium sample to PC gas, and no changes of the surface could be detected after exposure. The results of this investigation are included in Supporting Note 2. From these, we concluded that the Li used in this study was already passivated and did not react further with PC. This is already stated in the manuscript in line 100: “From this, we conclude that the substrate surface is non-reactive (see sample preparation and supporting information S2).”

We also addressed the issue of possible contaminations from the substrate in the original manuscript now on page 5, line 106ff: “The absence of signal in the Li 1s region supports the visual assessment that the drop of solvent at the measurement spot is thick enough to prevent any signal from the underlying substrate. The absence of a Li peak also shows, that there is no dissolution of Li from the substrate into the solvent.” To clarify this point we modified the text.

Since no trace of lithium (or other dissolved compounds) could be detected on the surface of the pure PC drop, we interpret this as the compounds present on the lithium surface cannot be dissolved in the solvent/electrolyte. Taken both points, droplet thickness and lack of signal from lithium or dissolved compounds, into account, we concluded that we could fully omit the influence of the substrate in the following analysis.

Changes to the manuscript

p. 5 line 109: “Thus, it is concluded that only PC contributes to the signal and that the substrate does not contaminate the surface of the drop.”

3. *Authors detected LiF and CF₃ peak in F1s, but surprisingly no CF₂, or CF peaks (which are intermediate to form LiF).*

Author reply:

Different C-F_x compounds are expected to be found between 287.5 eV and 293 eV in the C1s spectra depending on the number of fluorine atoms in the C-F_x group. In the same binding energy region various carbon-oxygen configurations can also contribute to the intensity. This means that the peaks from these compounds may overlap, making them hard to distinguish. We indeed see two additional peaks in the C 1s spectra of the electrolyte drop that do not correspond to the solvent or intact salt. These could be C-F₃ decomposition products in the form of C-F_x. To clarify the discussion surrounding the C1s spectra we have added a sentence on this topic, updated Table 1, and added two references in relation to the discussion surrounding Figure 2. In the manuscript we notice that we have salt decomposition, however due to the difficulty in separating overlapping contributions the experimental data does not allow for a detailed discussion.

Changes to the manuscript:

Table 1: updated with C-F_x

p. 5 line 114: "...expected for a carbon in carbonyl group or a carbon bound to a single fluorine atom."

p. 7 line 152: "...a peak at approximately the carbonyl/C-F binding energy (288 eV) is observed also for the Elect-Drop sample. In addition, a further peak at 289.4 eV is seen that could be explained by a decomposition of the CF₃ group from the TFSI anion to different C-F_x species (see also Figure 2)."

p. 9 line 216: "The determination of the exact decomposition mechanism of the salt is beyond the scope of this article but agrees with the previously postulated LiTFSI decomposition routes [41, 42] and reported (electro)chemical instability of Mg₂TFSI [30]."

4. LiF is insoluble in PC solvent, but the presence of LiF peak in F1s spectra indicates, authors are observing the metal-droplet interface?

Author reply:

Concerning the probing depth and the possibility of observing the metal-droplet interface, we refer to the general reply and the related changes, conveying that we cannot see the metal-droplet interface.

However, we are thankful for the reviewer comment since it showed that our description of the peak assignments in the F 1s can be misleading. We do not believe that we have LiF in solution but wanted to point out that the binding energy we observe matches to LiF and we therefore believe to observe solvated Li⁺ and F⁻ ions. The F⁻ ions would in this case stem from salt degradation, and the rather poor solvability according to Jones et al. (<https://www.sciencedirect.com/science/article/pii/S0378381209002957?via%3Dihub>) could then be a reason for the accumulation of both Li⁺ and F⁻ ions at the droplet surface. We have clarified this in the manuscript.

In addition, we updated parts of the supporting information to further clarify that we observe solvated ions and not LiF.

Changes to the manuscript:

p. 9 line 206: "The CF₃ chemical environment is expected at the higher binding energy position (688.4 eV). The second component has a binding energy of 684.7 eV and most likely originates from solvated fluoride ions (based on the general assignment of this binding energy to LiF for solid samples)."

p. 10 line 236: "Combining the Li intensity with the observed fluoride ion peak intensity at a binding energy of 684.7 eV, we evaluated to what extent F⁻ ions could compensate for the excess Li."

Changes to the Supporting Information:

Page 6 (S6): "In order to evaluate if F⁻ can act as a counter-anion for the clear intensity increase in ionized lithium at the droplet surface, the relative intensity contribution from F⁻ (seen at a binding energy of 684.7 eV) to the Li 1s spectrum is evaluated."

Page 6 (S6): "From these calculations, we estimate that F⁻ can act as counter-anion to roughly 70% of the total Li⁺ and TFSI to roughly 10%."

5. *small amount of elemental sulfur (~163eV) in S2p spectrum is interesting, but again, looks more like contaminant issue., although author claims this is not focus of the manuscript, these critical observations need to be analyzed (at least provided hypothetical explanations).*

6. *Authors observe decomposition of CF3 (as LiF) and SO2 (as elemental sulfur) but no extra peaks observed in Nitrogen spectra?., If the TFSI molecule is decomposing, isn't Nitrogen from TFSI will also form byproducts?*

Author reply:

In our view, the comments 5 and 6 are closely related since they deal with the degradation of the TFSI anion. We therefore address the raised questions from these two comments in the following.

All spectra probing elements of the TFSI ion show signs of salt degradation; fluorine products have been discussed above (see answer to comments 3 and 4), and sulfur degradation products are seen in the form of a second peak at lower binding energy in the S 2p spectrum. For nitrogen the degradation may explain the asymmetric N 1s peak shape. The N 1s peak is however generally a broad peak and in comparison to other peaks, it has a low PES cross-section. This results in relatively noisy data and close lying peaks are generally difficult to resolve univocally. Thus, the peak is instead curve-fitted using one peak representing all nitrogen containing compounds. A fitting using multiple peaks could be done in many ways, and thus this would give no conclusive result, but could rather be misleading.

Since no surface contaminants are seen on the solvent sample and the droplet is much thicker than the probing depth, we assign these contributions to salt decomposition rather than surface contaminants.

Changes to the manuscript:

The following sentence and references have been added for clarity.

p. 10 line 214: “Additionally, the asymmetric shape of the N 1s peak further supports that some of the TFSI-ions have degraded.”

p. 10 line 216: “The determination of the exact decomposition mechanism of the salt is beyond the scope of this article but agrees with the previously postulated LiTFSI decomposition routes [41, 42] and reported (electro)chemical instability of Mg₂TFSI [30].”

p. 10 line 219: It is based on the intensities that can be clearly-attributed to the intact LiTFSI salt (i.e., not including intensity that clearly can be attributed to possible-decomposition products).

7. *Authors need to identify and discuss more about C1s peak around 291eV (is this CF2/CF)?*

Author reply:

The peak that the reviewer is referring to at 291 eV is the CO₃ peak of the PC solvent molecule, which has been discussed in relation to Figure 1 where the C 1s and O 1s spectra of the solvent and electrolyte are discussed. This CO₃-peak could possibly overlap with some CF₂ compounds, but in view of the relative intensity with the other C1s solvent peaks, most of the intensity around 291 eV stems from the PC solvent.

Changes to the manuscript:

In Figure 2, all PC components are now marked with dotted lines for CH (=C1) and CO₃ (=C3) and a dashed line for the ether groups (C2), since the latter was used for the evaluation. Additionally, the following explanation was included:

p. 7 line 158: “...to separate the different contributions of solvent (PC C1, C2, and C3 according to inset in Fig. 1) and salt.”

8. *Another concern is the results vary with different measurement spots, there is no convincing results for such an inhomogeneous sample/measurement conditions.*

Author reply:

As seen in the carbon spectra of Figure 2 the overall line shape is very similar for the 5 different measurement spots, with only some small variations in the hydrocarbon intensity that could be due to adsorption of ambient hydrocarbons during the measurements. If the reviewer is referring to the different intensity levels, this was due to the physical constraints in sample positioning with respect to the analyzer in the SPECIES end station. If we moved the sample too far from this optimum spot, the overall intensity dropped significantly. Nevertheless, the relative intensities measured at the same spot was not influenced by this, and this was continuously taken care of during data analysis.

Reviewer #3 (Remarks to the Author):

Reviewer report on:

Manuscript Number: Nature Comm.

NCOMMS-18-33746

Authors: Maibach et al.

Title: The surface of a battery electrolyte drop seen through ambient pressure x-ray photoelectron spectroscopy

General:

At first a personal comment: I am not a regular reader of "Nature Comm.". After inspecting some of the papers in the current field, I feel that most of the published papers have a more general message than the present one. This is my main doubt and thus you should include your editorial position and the opinion of the other reviewers into the final decision.

Nevertheless, the paper is dealing with a very actual challenging field: the improvement of the possibilities for storage of electrical energy efficiently in batteries for future use of renewable energy.

The characterization of the complex chemical processes in the batteries is for that aim of existential importance. For more realistic analysis the closure of the "pressure gap" is of main interest. The experiments of the authors here are really at the forefront in the community.

The experiments, results and conclusions are convincing (with some exceptions - see remarks). To learn about the electrolyte/salt behavior is of high importance and the concept of the authors is valid. Main result is that XP measurements at liquid PC-based electrolyte is reliable possible and that demixing of electrolyte components can be proved. The study is a prerequisite for real electrochemical in-operando experiments, but it is still some steps away from it. And, I feel, in the "background" section the reader is given a little bit too much hope for such studies.

Author reply:

We believe that our results and conclusions are of general interest to a broad research community in physics, chemistry, and biology with a common interest in solid/liquid (e.g. corrosion, biological cell membranes) and liquid/gas (e.g. aerosols in atmosphere, gas sensors) interface phenomena. Probing such interfaces using APPES always first requires understanding the liquid/gas interface for a correct data interpretation. As a general trend science and research is moving towards measurements in more realistic environments and our results make a clear contribution to this broad field. Our results also clearly show that operando studies using a highly surface sensitive technique are now a realistic target if stable conditions for the liquid phase are achieved.

In addition, being able to use complex liquids such as battery electrolytes is an important achievement by itself since previous approaches to characterize liquids in PES such as liquid-jet experiments could not tolerate the relatively high salt concentration and low viscosity of these electrolytes. We are also convinced that the true operando APPES measurements of the battery electrodes are coming shortly, as very promising experimental advancements have already been taken by us and others in this area. The steps taken in the current article are important for the methodology development and are a prerequisite for the correct interpretations of future, truly operando APPES data. We therefore are convinced that our results will appeal to the broad readership of Nature Communications.

To pinpoint the topic of this article, being an important methodology development towards operando APPES measurements of battery electrodes, we have shortened the background on operando APPES and sharpened the text accordingly. Specifically, the following changes were made:

Changes to the manuscript:

p. 4, line 73: removed possible, included “a realistic target”

p. 4, line 73: removed sentence: This means that the surface sensitive AP-XPS measurements can be performed simultaneously with the electrochemical characterization.

p. 4 line 76: Sentence “Recognizing the opportunity with new operando AP-XPS measurements, one should keep in mind that the method is still in its cradle and thorough methodology development is necessary for correct interpretation of the results.” Has been changed to: “Recognizing the prospect of future operando APPEES measurements, we here identify and take a necessary step in the required methodology development.”

p. 4 line 79: removed already

p. 4 line 82: removed most likely

p. 4 line 83: included “as the top electrolyte layer will dominate the signal”

Detailed remarks:

If you decide for publication I have to add some specific remarks:

1. I am also not a native English speaker, but - in some parts the English has to be improved! One example - last sentence of conclusions: “Only if one understands how the liquid electrolyte behaves under AP-XPS condition can a correct interpretation of in-situ and operando battery experiments be achieved.”

Author reply:

We have consulted an English native speaker and revised the text accordingly. Specifically, the above-mentioned sentence has been changed.

Changes to the manuscript

p. 11, line 271: “A correct interpretation of in-situ and operando APPEES battery experiments can only be achieved when we fully understand the behavior of the liquid electrolyte under the given conditions.”

2. All abbreviations have to be defined? In the abstract already or in the main text? E.g. AP-XPS is explained in line 2, not in line 47, once more explained in line 57

Author reply/Changes to the manuscript:

We have carefully gone through the manuscript and defined all abbreviations. We have chosen to define relevant abbreviations once in the abstract, main text and the conclusions (since these parts could be read separately).

3. The paper should be clearly readable also for non-expert readers, the mixture of PES and XPS here is not really necessary.

Author reply/Changes to the manuscript:

This is a good point. In view of the existence of two equally common abbreviations of ambient pressure (X-ray) photoelectron spectroscopy being AP-XPS and APPEES, we exchange AP-XPS with APPEES throughout the article for consistency.

4. line 19: fossil-free “society” - what is this...?

Author reply:

We thank the reviewer for finding this error. This should read fossil fuel-free society. This has been updated in the article.

Changes to the manuscript:

p. 3 line 19: fossil fuel-free society

5. l 63: what means “tubing” here

Changes to the manuscript:

p. 4 line 68: For clarity the phrasing “tubing” has been exchanged to “dedicated tubes/pipes for electrolyte inlets”.

6. + [33] if I understood the paper right there Maibach et al. already measured XP spectra of a drop of PC - thus it is not really the “first time” here...

Author reply:

In our previous study cited as Ref. 33 in this manuscript, we were not able to characterize pure propylene carbonate (PC) with APPEs. Instead we only measured on the electrolyte 1M LiClO₄ in PC. While the previous study also used PC as a solvent, the data presented in this manuscript to the best of our knowledge is the first time a liquid of pure propylene carbonate (without any salt) was measured. Previous attempts on this have been unsuccessful, as the solvent has consistently been pumped away.

7. l 65: 1 M concentration - of what

Author reply:

We thank the reviewer for finding this error.

Changes to the manuscript:

p. 4 line 70: “a realistic battery electrolyte at 1 M salt concentration”

8. l 67-70: “using a set-up with dedicated electrical feedthroughs make operando AP-XPS measurements now possible” – this makes the reader wrong hope, not shown in the current paper!

Author reply:

We do not wish to mislead the reader and thus we have clarified the text.

Changes to the manuscript:

p. 4 line 72. “Maintaining a liquid electrolyte in the analysis chamber and using a set-up with dedicated electrical feedthroughs make operando APPEs measurements now a realistic target”

9. l 80-81: “and the depth distribution of its components” - not really “depth distribution” - hints for demixing!

Changes to the manuscript:

p. 4 line 87: For clarity we changed the phrasing to read “concentration variations”.

10. l 92 - comment that the “pristine” surface is Li-carbonate not Li metal !

Author reply:

For this comment we refer to the answer to reviewer 1:

Prior to depositing the electrolyte drop, the lithium metal substrate was thoroughly investigated using PES. It was found that the pristine Li sample surface consisted of Li₂CO₃ and other compounds, and only a minor metallic Li 1s peak from the bulk lithium metal could be observed. The inertness of this lithium substrate was checked by exposing the lithium sample to PC gas, and no changes of the surface could be detected after exposure. The results of this investigation are included in Supporting Note 2. From these, we concluded that the Li used in this study was already passivated and did not react further with PC. This is already

stated in the manuscript in line 100: “From this, we conclude that the substrate surface is non-reactive (see sample preparation and supporting information S2).”

11. *from here energy values are discussed... a hint that the BE scale is calibrated (C1s 285 eV - see “data treatment”) would here be helpful*

Changes to the manuscript:

p. 5 line 104: We added the sentence: “The binding energy scale is calibrated versus the hydrocarbon peak at 285 eV (further information in data treatment section).”

12. l 108: “these are the first spectra of PC in liquid phase.” ??? [33] see comment above

Author reply:

We include here the same answer as above: Previously we have measured the electrolyte 1M LiClO₄ in propylene carbonate. This is the first time pure propylene carbonate was measured as a liquid.

13. l 133, 140: 284.9 eV ??? BEs are calibrated to 285.0 eV!

Author reply/changes to the manuscript:

We thank the reviewer for finding this error. This value has been updated to 285.0 eV.

14. *concentration calculation - (S5) here a comment on the used procedure would be helpful..., error discussion?*

Author reply:

We thank the reviewer for providing this valuable feedback also on the supporting information.

We have included a discussion of the used estimation procedure in the Supporting Note 5 and provided further information in Supporting Note 3. The latter concerns the evaluation of the salt to solvent ratio and since here, only C1s peaks are compared, the errors in this evaluation are significantly lower.

Changes to the Supporting Information:

Page 5 (S5): When evaluating the salt stoichiometry only peaks that could be identified as stemming from the intact salt were included. This means that some elements (N, Li) could be overestimated since possible degradation products would still appear at similar binding energies. For S, C and F distinguishable chemical environments can be seen, allowing us to omit these contributions for the calculations. For oxygen further difficulties occur to exactly determine the salt content since the peak is overlapping with the solvent peak.

Keeping these difficulties in mind, we acknowledge that we can have rather large variations in the calculated ratios, also due to some uncertainties in the assumptions made to estimate σ_i and λ_i [3,5]. Therefore, we refrain from drawing conclusions for deviations on the scale up to ~50 %, but note that the lithium content exceeds all other variations by far.

Page 3 (S3): Since these calculations are performed for peaks of the same element in the same spectra intensity variations due to the spectrometer, analyser, sample setup and cross-sections are eliminated. Thus, the only parameter necessary for the evaluation is the peak area. This value could vary slightly depending on the peak fitting if there are several overlapping peaks. Therefore, only the C2 component of PC at 287 eV was considered and not the C3 component at 290.6 eV corresponding to the carbonate moiety in PC, since the latter might be overlapping with CF_x components. The C-O and CF₃ peaks are well separated from other contributions and therefore we believe the results to be reliable.

15. l 176 177: “fresh location” - what this really means for a liquid droplet? I would assume always convection of material, at least at the surface...

Author reply:

Our results clearly show that convection/diffusion of degradation products are slow enough as not to influence a nearby fresh spot within the time frame of the measurements. This is based on the observation that no significant signs of degradation products can be seen for repeated C 1s spectra measured at different fresh spots, while spectra obtained when returning to a previously measured spot show clear signs of degradation products. However, we agree with the reviewer that there should be some degree of convection/diffusion within a liquid drop. From our results in this study, we can conclude that in this case the total concentration of degradation products is smaller than the detection limit of the measurements. To clarify this point, we added the following sentence to the article.

Changes to the manuscript:

p 9, line 197: “Finding such a homogeneous electrolyte surface shows that decomposition products, observed during exposure to radiation (see S4), do not spread to nearby measurement spots within the time frame of the measurements.”

16. l 195 very noisy peak !

Author reply:

Although the quality of the data in this manuscript is much better than the general APPES measurements, still some of the data has a lower signal to noise ratio. This is partly related to the instrumental constraints of the SPECIES end station where the sample position cannot be efficiently optimized at all points available at the liquid drop. However, this does not affect the conclusions since for all measurements we always have control over relative intensities. To clarify this, we added a sentence to the manuscript:

Changes to the manuscript

p. 9 line 199: “The variations in signal-to-noise-ratio are due to the instrument constraints, where not all sample positions can be fully optimized with respect to maximum count rate and the available measurement time.”

17. l 208, l 217: table 2, discussion for C: Li = 2 : 12, mandatory! I cannot follow the discussion with mainly formation of LiF !! There is not enough excess of fluorine!

Author reply:

As written in the article, only the intensity stemming from the **intact** LiTFSI is considered for evaluating the stoichiometry, as visualized with filled grey peaks in figure 2 (see lines 244, 245, and 247).

For lithium though, all Li⁺ ions will appear at the same binding energy, regardless of any salt decomposition and in this case all Li⁺ ions are therefore included in the calculations. The fluoride ions believed to charge compensate for some of the excess Li⁺ are on the other hand clearly separated from the intact salt peak and therefore not included in the calculations of the salt stoichiometry.

When integrating the area of the non-shaded F 1s peak at 684.7 eV it is found that the amount of F⁻ could counter 70 % of the Li⁺ ions. With TFSI⁻ countering an additional 10 % of Li⁺ ions this leaves 20 % Li⁺ with an unknown counter ion. It is suggested that this intensity can stem from lithium carbonate compounds (see line 242). Details on these calculations can also be found in supporting note 6.

18. l 222-223: “The remaining intensity could stem from lithium carbonate type compounds and related solvated Li ions” - please, clearer statement on this
Proposal: what about making the same experiment with a substrate other than a Li foil? So one can rule out dissolution of Li...

Author reply:

We agree that for many reasons, it would be very interesting to perform the same experiments on a gold substrate. However, for the results presented in this article we are convinced that the substrate does not influence the results. We refer to the general remarks in the beginning and also to the answer given to reviewer 1:

Prior to depositing the electrolyte drop, the lithium metal substrate was thoroughly investigated using PES. It was found that the pristine Li sample surface consisted of Li_2CO_3 and other compounds, and only a minor metallic Li 1s peak from the bulk lithium metal could be observed. The inertness of this lithium substrate was checked by exposing the lithium sample to PC gas, and no changes of the surface could be detected after exposure. The results of this investigation are included in Supporting Note 2. From these, we concluded that the Li used in this study was already passivated and did not react further with PC. This is already stated in the manuscript in line 100: “From this, we conclude that the substrate surface is non-reactive (see sample preparation and supporting information S2).”

We also addressed the issue of possible contaminations from the substrate in the original manuscript now on page 5, line 106ff: “The absence of signal in the Li 1s region supports the visual assessment that the drop of solvent at the measurement spot is thick enough to prevent any signal from the underlying substrate. The absence of a Li peak also shows, that there is no dissolution of Li from the substrate into the solvent.” To clarify this point we modified the text.

Since no trace of lithium (or other dissolved compounds) could be detected on the surface of the pure PC drop, we interpret this as the compounds present on the lithium surface cannot be dissolved in the solvent/electrolyte. Taken both points, droplet thickness and lack of signal from lithium or dissolved compounds, into account, we concluded that we could fully omit the influence of the substrate in the following analysis.

19. l 240: “salt enrichment” OK, but not “depth distribution” as mentioned earlier in the paper - please, discuss HAXPES advantages here, too

Author reply:

We agree with the reviewer that it would be beneficial to perform these types of APPES measurements using also higher photon energies, i.e. HAXPES. A higher photon energy would make the electrolyte surface less dominant. If the ultimate target is reaching operando APPES measurements of battery electrodes, this will also involve probing the buried interface between the solid electrode and the electrolyte. Here, HAXPES can be helpful in probing through the liquid phase to the buried interface. However, there is a trade-off between higher probing depth and interface sensitivity, and this would be an interesting topic for a following paper. The topic of synchrotron depth profiling is lifted in the background and a sentence was added on this in the conclusion.

Changes to the manuscript:

p. 3 line 50: “energy ranging from soft to hard X-rays [11, 12]”

p. 11, line 265: “Also, optimal conditions for probing a buried interface require understanding of the tradeoff between interface sensitivity and probing depth.”

20. l 264-266: what species are outgassing, water?,

how one can reach 2×10^{-3} mbar, if the vapor pressure of PC is 0.2 mbar; is there no evaporation of PC ???

Author reply:

We thank the reviewer for finding this unclarity. In the experimental preparations we used a rough vacuum pump that was able to reach a base pressure of 2×10^{-3} mbar. The experimental procedure is updated accordingly.

Changes to the manuscript:

p. 12 line 285: “Additionally, 10 ml PC were degassed by placing PC in a vessel with a small opening in a vacuum chamber (pressure approx. 2×10^{-3} mbar, using a backing pump).”

21. 1290... please, explain the geometry (size) of the droplet

Author reply:

We have included a schematic representation in figure 1a that clearly shows the sample (drop). For more information, we refer to the general remarks in the beginning of this document.

REVIEWERS' COMMENTS:

Reviewer #2 (Remarks to the Author):

Authors explained the results and answered the queries in satisfactory manner. I am happy with their responses and modifications to the manuscript. I recommend the publication of this article in the modified format.

Reviewer #3 (Remarks to the Author):

Dear authors, I am now satisfied with your contribution!